# Calnexin Is Involved in Forskolin-Induced Syncytialization in Cytotrophoblast Model BeWo Cells

**DOI:** 10.3390/biom12081050

**Published:** 2022-07-28

**Authors:** Hitomi Matsukawa, Midori Ikezaki, Kaho Nishioka, Naoyuki Iwahashi, Masakazu Fujimoto, Kazuchika Nishitsuji, Yoshito Ihara, Kazuhiko Ino

**Affiliations:** 1Department of Obstetrics and Gynecology, School of Medicine, Wakayama Medical University, Wakayama 641-8509, Japan; hitomi_m@wakayama-med.ac.jp (H.M.); ka-matsu@wakayama-med.ac.jp (K.N.); naoyuki@wakayama-med.ac.jp (N.I.); kazuino@wakayama-med.ac.jp (K.I.); 2Department of Biochemistry, School of Medicine, Wakayama Medical University, Wakayama 641-8509, Japan; ikezaki@wakayama-med.ac.jp (M.I.); nishit@wakayama-med.ac.jp (K.N.); 3Department of Diagnostic Pathology, Kyoto University, Kyoto 606-8507, Japan; fujimasa@kuhp.kyoto-u.ac.jp

**Keywords:** β-hCG, calnexin, chaperone, placenta, syncytialization, trophoblast

## Abstract

Calnexin (CNX), a membrane-bound molecular chaperone, is involved in protein folding and quality control of nascent glycoproteins in the endoplasmic reticulum. We previously suggested critical roles of calreticulin, a functional paralogue of CNX, in placentation, including invasion of extravillous trophoblasts and syncytialization of cytotrophoblasts. However, the roles of CNX in placentation are unclear. In human choriocarcinoma BeWo cells, which serve as an experimental model of syncytialization, CNX knockdown suppressed forskolin-induced cell fusion and β-human chorionic gonadotropin (β-hCG) induction. Cell-surface luteinizing hormone/chorionic gonadotropin receptor, a β-hCG receptor, was significantly down-regulated in CNX-knockdown cells, which suggested the presence of a dysfunctional autocrine loop of β-hCG up-regulation. In this study, we also found abundant CNX expression in normal human placentas. Collectively, our results revealed the critical role of CNX in the syncytialization-related signaling in a villous trophoblast model and suggest a link between CNX expression and placenta development.

## 1. Introduction

Formation of a placenta is critical for pregnancy and fetal health in humans. During the first few weeks of gestation, the placenta rapidly develops from embryonic cells. After implantation of the embryo, three types of trophoblasts in the placenta—cytotrophoblasts (CTBs), syncytiotrophoblasts (STBs), and extravillous trophoblasts—are produced for critical functions in placental development [1]. CTBs and STBs form villous trophoblasts that provide nutrients to the embryo and make up part of the placenta. STBs are multinuclear cells formed by the fusion of CTBs. The differentiation of CTBs into STBs, that is to say, syncytialization is an essential process for maintenance of a successful pregnancy and depends on multiple elements including environmental factors and cytokines such as human chorionic gonadotropin (hCG) and human placental lactogen [2]. During the course of syncytialization, hCG is mainly produced by placental STBs, and hCG promotes syncytialization in an autocrine/paracrine manner, through the binding of hCG to the receptor luteinizing hormone/chorionic gonadotropin receptor (LHCGR) [3,4,5]. Inadequate fusion of CTBs may lead to placental dysfunction and pregnancy complications such as preeclampsia (PE) and fetal growth restriction (FGR) [6]. PE and FGR are the most serious pregnancy complications and can be fatal to both mothers and infants. To date, definitive treatment of PE is delivery of the infant or termination of the pregnancy [7,8]. 

The endoplasmic reticulum (ER) is an important compartment for proper protein folding and has a sophisticated quality control system to eliminate improperly folded proteins during protein secretion [9,10]. Protein folding is assisted by molecular chaperones that recognize misfolded or unfolded proteins and sequester them in the ubiquitin-proteasome system for elimination. ER stress has been implicated in placental dysfunction in human pregnancy complications such as PE and FGR, which confirms the importance of the ER and its protein quality control system in placental development [11,12].

Calreticulin (CRT) is a soluble ER-resident molecular chaperone which facilitates proper folding of nascent glycoproteins and regulates Ca^2+^ homeostasis [13]. CRT is expressed in the human placenta [14,15,16,17], and it has been reported that the level of CRT is higher in maternal blood during pregnancy than in nonpregnant women [15,17]. There have been some reports about the role of CRT in trophoblast invasion [16,18]. We also reported that CRT was involved in the invasion of extravillous trophoblasts by regulating *N*-glycosylation of integrins and adhesion to extracellular matrix proteins [19]. CRT also had a critical role in trophoblast syncytialization by regulating membrane expression of E-cadherin and E-cadherin-mediated cell-cell interactions [20]. These results suggest the crucial importance of the ER chaperone CRT in placental development.

Calnexin (CNX) is a type I membrane-bound ER chaperone and a paralog protein of CRT in the ER [21]. CNX and CRT are preferentially involved in *N*-glycoprotein biosynthesis in the ER. Both chaperones ensure the proper folding of nascent or unfolded polypeptides, mainly via interactions with monoglucosylated *N*-glycans on substrate proteins [22,23,24]. They work cooperatively or separately on quality control of *N*-glycoproteins in the ER, which is a cycle named the CNX/CRT chaperone cycle [25,26]. CNX and CRT take part in various physiological processes via chaperone functions to control maturation and sorting of target proteins, but the particular functions of both chaperones are not yet fully clarified. CNX is reportedly involved in various cellular events related to neural differentiation and development [27,28], as well as inflammation and immunity [29]. In humans, CNX is expressed in most organs and tissues including the placenta (Human Protein Atlas, https://proteinatlas.org, accessed on 25 July 2022.), which strongly suggests that CNX may also participate in placental development. To date, however, the physiological importance of CNX in the placenta, unlike that of CRT, has not been fully elucidated.

In the present study, we focused on the roles of CNX in placental development and investigated effects of CNX down-regulation on trophoblast differentiation in vitro by using forskolin (FSK)-induced syncytialization of BeWo cells, which we used as a model of CTBs, and also examined the expression of CNX in human placentas.

## 2. Materials and Methods

### 2.1. Materials

We acquired a rabbit polyclonal anti-CNX (C-terminal) antibody (Cat No. SPA-860), rabbit polyclonal anti-CRT antibody (Cat No. SPA-600), rabbit polyclonal anti-BiP antibody (Cat No. SPA-826), and mouse monoclonal anti-PDI antibody (Cat No. SPA-891) from Stressgen (San Diego, CA, USA). A mouse monoclonal anti-β-actin antibody was acquired from Santa Cruz Biotechnology (Cat No. sc-1616, Dallas, TX, USA). We obtained a rabbit polyclonal anti-CNX (N-terminal) antibody (Cat No. ADI-SPA-865), and mouse monoclonal anti-E-cadherin antibody (Cat No. 610181) from Enzo Life Sciences (New York, NY, USA), and BD Biosciences (Franklin Lakes, NJ, USA), respectively. Rabbit polyclonal and mouse monoclonal anti-β-hCG antibodies were purchased from Proteintech (Cat No. 11615-1-AP, Rosemont, IL, USA) and Abcam (Cat No. ab9582, Cambridge, UK), respectively. Rabbit polyclonal and mouse monoclonal anti-luteinizing hormone/chorionic gonadotropin receptor (LHCGR) antibodies were obtained from Thermo Fisher Scientific (Cat No. PA5-97923, Waltham, MA, USA) and Novus Biologicals (Cat No. NBP2-53726, Centennial, CO, USA), respectively. Horseradish peroxidase (HRP)-conjugated secondary antibodies were from Dako (Cat No. P0448 and P0260, Glostrup, Denmark). All other reagents used in this study were of high grade and were obtained from Sigma-Aldrich and Wako Pure Chemicals (Osaka, Japan).

### 2.2. Collection of Human Tissue Samples and Preparation of These Samples

Individual patients gave their informed consent for use of the specimens of placentas. Samples of third-trimester (at 34–39 weeks) human placentas were collected at cesarean sections that were performed before labor started. The Ethics Committee of Wakayama Medical University approved this study (authorization number: 1690).

### 2.3. Immunohistochemical Analysis

Paraffin-embedded blocks of placental tissues were cut into 3-µm-thick sections, deparaffinized, and rehydrated, after which endogenous peroxidase activity was blocked with 0.3% hydrogen peroxide in methanol. For antigen retrieval, these sections were heated in 1 mM EDTA (pH 8.0) in a pressure cooker for 10 min. The sections were then incubated with rabbit polyclonal anti-CNX (C-terminal) antibody, followed by incubation with HRP-conjugated secondary antibody. We detected signals by using the Histofine Simple Stain MAX PO reagent (Nichirei Biosciences, Tokyo, Japan) and 3,3ʹ-diaminobenzidine as the substrate.

### 2.4. Culture of Cells

Human choriocarcinoma BeWo cells are used extensively in models of trophoblast differentiation and syncytialization [20,30]. We purchased these cells from the American Type Culture Collection (Manassas, VA, USA), and the cells were authenticated by the Japanese Collection of Research Bioresources Cell Bank (National Institute of Biomedical Innovation Japan, report no. KBN0410). We grew these cells in RPMI 1640 medium (Wako Pure Chemicals) with added supplementation of 10% fetal calf serum (BioWest, Nuaillé, France) and penicillin, streptomycin, and amphotericin B (Life Technologies, Carlsbad, CA, USA) at 37 °C in 5% CO_2_ and 95% air.

### 2.5. Creation of Stable CNX-Knockdown Cells

We purchased OriGene pRS shRNA vectors containing short hairpin RNA for CNX (CNX-shRNA, #TR314210) and non-specific shRNA (#TR20003), and the puromycin-resistant marker from OriGene Technologies (Rockville, MD, USA). We transfected the BeWo cells with the vectors by using Lipofectamine 2000 Reagent (Invitrogen, Carlsbad, CA, USA), and we acquired stably gene-transfected cells, in which the shRNA is integrated, by using puromycin (Wako Pure Chemicals, 1.0 µg/mL) for selection. We isolated two CNX-knockdown clones (CNX-shRNA-1 and CNX-shRNA-2), in which expression of CNX protein was suppressed, from the transfectants.

### 2.6. Analysis of Immunoblots

Cells were harvested and lysed in Lysis Buffer A (10 mM Tris-HCl [pH 7.5], 150 mM NaCl, 1% Nonidet P-40) that contained the following protease inhibitors: 4 mM Pefabloc, 1 µM pepstatin, 1 µM leupeptin, and 200 µM phenylmethylsulfonyl fluoride (Roche, Basel, Switzerland). We centrifuged the lysates at 10,000× *g* for 10 min at 4 °C, and we collected the supernatants and separated them by means of 10% sodium dodecyl sulfate (SDS)-polyacrylamide gel electrophoresis. Proteins were then transferred to polyvinylidene difluoride membranes (Immobilon-P, Merck Millipore, Burlington, MA, USA). Membranes were blocked with 5% skim milk (BD Biosciences) and 0.05% Tween 20 (Wako Pure Chemicals) in Tris-buffered saline (pH 7.6) and were incubated with the following primary antibodies: anti-CNX antibody (N-terminal), 1:5000; anti-β-actin antibody, 1:1000; anti-BiP antibody, 1:5000; anti-CRT antibody, 1:10,000; anti-E-cadherin antibody, 1:3000; rabbit polyclonal anti-β-hCG antibody, 1:2000; and mouse monoclonal anti-LHCGR antibody, 1:1000; incubation with HRP-conjugated secondary antibodies followed. Signals were identified by using the Immobilon Western Chemiluminescent HRP substrate (Merck Millipore) and were quantified means of by densitometry with ImageJ version 1.52a (National Institutes of Health, Bethesda, MD, USA). β-Actin was used as a loading control. For cell-based assays, we treated BeWo cells and their variants with FSK (50 µM, Sigma-Aldrich) for 72 h, after which we lysed them in Lysis Buffer A. Conditioned media were also harvested, and β-hCG protein levels were analyzed by means of Western blotting as described above.

### 2.7. Reverse Transcription-Polymerase Chain Reaction (RT-PCR) and Real-Time Quantitative RT-PCR (RT-qPCR)

We used the TRIzol reagent (Thermo Fisher Scientific) to obtain total cellular RNA. We carried out RT-qPCR with the CFX96 Touch Real-Time system (Bio-Rad Laboratories, Hercules, CA, USA) and the iTaq Universal SYBR Green One-Step Kit (Bio-Rad Laboratories). Primer oligonucleotide sequences used for RT-qPCR amplification were as follows: β-hCG (GenBank accession number J00117.1), forward: 5’-CTACTGCCCCACCATGACCC-3´, reverse: 5´-TGGACTCGAAGCGCACATC-3´; and GAPDH (GenBank accession number M33197), forward: 5´-GAGTCAACGGATTTGGTCGT-3´, reverse: 5´-GACAAGCTTCCCGTTCTCAG-3´. We performed all RT-qPCR experiments in triplicate. We processed the data with Bio-Rad CFX Manager version 3.1 (Bio-Rad Laboratories, Berkeley, CA, USA), and we calculated expression levels by using the comparative ΔΔCt method, with GAPDH as the reference RNA.

### 2.8. Analysis of Cell Fusion

We used coverslips to grow cells in six-well culture plates, with cells cultured for 24 h. We then treated the cells with FSK (50 µM) or 0.05% dimethyl sulfoxide (DMSO) for 24, 48, or 72 h. Cells on coverslips were fixed with 4% paraformaldehyde for 20 min at room temperature, followed by blocking and permeabilization with the Animal Free Blocker (Vector Laboratories, Burlingame, CA, USA) containing 0.05% saponin for 20 min at room temperature. Cells were then stained with mouse monoclonal anti-zonula occludens protein-1 (ZO-1) antibody (Cat No. ab216880, Abcam, 1:100) and Alexa Fluor 488-conjugated anti-mouse IgG (Jackson ImmunoResearch Labs, West Grove, PA, USA). Stained specimens were mounted with Vectashield Mounting Medium containing DAPI (Vector Laboratories) and examined with a LSM700 laser scanning confocal microscope and the LSM software ZEN 2012 (Carl Zeiss, Jena, Germany). For each group we counted the numbers of fused cells in 8–10 microscopic fields that were randomly selected. In each field, we counted both the total number of nuclei and the numbers of nuclei in STB-like fused cells. We then calculated the percentages of fused cells as the ratio of the number of nuclei in the fused cells to the total number of nuclei in the microscopic field.

### 2.9. Analysis of Cell Proliferation

We seeded 7.5 × 10^3^ cells in 96-well plates and cultured them for 24, 48, and 72 h, after which we fixed the cells with 4% paraformaldehyde in phosphate-buffered saline (PBS) for 20 min at room temperature and stained them with 0.01% crystal violet. We then lysed the cells with 100 µl of lysis buffer B (10% SDS plus 10 mM HCl), and we estimated the number of cells photometrically by measuring lysate absorbance at 595 nm with a 680XR Microplate Reader (BioRad, Hercules, CA, USA).

### 2.10. Analysis via Immunofluorescence Microscopy

We fixed cells grown on coverslips with 4% paraformaldehyde for 20 min at room temperature, after which cells were blocked and permeabilized with the Animal Free Blocker containing 0.05% saponin for 20 min at room temperature. The cells were then incubated with primary antibodies (anti-CNX antibody, 1:200; anti-E-cadherin antibody, 1:300; anti-β-hCG antibody, 1:100; rabbit monoclonal anti-LHCGR antibody, 1:100; and anti-PDI antibody, 1:300), followed by incubation with secondary antibodies conjugated to Alexa Fluor 488 or Alexa Fluor 555 (Thermo Fisher Scientific). Specimens were mounted with Vectashield Mounting Medium containing DAPI and examined with a LSM700 laser scanning confocal microscope.

### 2.11. Analysis of Cell-Surface LHCGR by Biotinylation

Cells grown overnight on coverslips were washed with ice-cold PBS, and cell-surface proteins were biotinylated by incubation of cells with 1.0 mg/mL EZ-Link Sulfo-NHS-SS-Biotin (Thermo Fisher Scientific) in PBS, with gentle rocking, for 30 min at 4 °C. We quenched the biotinylation reaction by using 100 mM Tris-HCl (pH 7.4) in ice-cold PBS. We then rinsed the cells with ice-cold PBS and lysed them in lysis buffer A. Biotinylated proteins were collected with avidin-conjugated agarose beads (Thermo Fisher Scientific), after which the beads were washed four times with lysis buffer A. Proteins were eluted in SDS sample buffer (0.125 M Tris-HCl, 4% [wt/vol] SDS, 20% [vol/vol] glycerol, and 0.01% [wt/vol] bromophenol blue) and were heated at 95 °C for 5 min. The samples then underwent immunoblot analysis with the anti-LHCGR antibody as described above.

### 2.12. Preparation of β-hCG-Containing Conditioned Medium

Cells were treated for 48 h at 37 °C in Opti-MEM with or without FSK (50 µM). Conditioned media were collected, and low-molecular-weight chemicals, such as FSK, were removed from the media by using an Amicon Ultra centrifugal filter (10-kDa cutoff) (Millipore, UFC5003BK) according to the manufacturer’s protocol. Conditioned medium samples (0.5 mL) were diluted 18-fold in RPMI 1640 containing 10% FCS, in which the total volume was adjusted to 9.0 mL. FSK-induced production and secretion of β-hCG in the conditioned medium samples was confirmed by immunoblot analysis using the specific antibody against β-hCG. The conditioned medium samples were used as FSK-treated CM that contains β-hCG or Untreated CM. For stimulation, cells were cultured overnight in 6-well plate, washed with PBS, then treated for 48 h in 1.5 mL of FSK-treated CM or Untreated CM.

### 2.13. Statistical Analysis

Data are shown as means ± SDs. We used Dunnett’s multiple comparisons test or unpaired Student’s *t*-test to analyze the data, with *p* values of <0.05 said to be significant.

## 3. Results

### 3.1. Establishment of Stable CNX-Knockdown BeWo Cells

The expression of Collection of Human Tissue Samples and Preparation of These Samples was investigated in human placental tissues. Immunohistochemical analysis of human placental tissues from the third trimester revealed that STBs in outer layers of the villi and CTBs in the inner layers expressed CNX, and we also observed CNX-expressing cells in villous stroma (Figure 1a and Appendix A). Specificity of the anti-CNX antibody used was confirmed by immunostaining with an isotype control IgG (Appendix A). Because we previously reported the critical role of CRT in syncytialization of BeWo cells [20], we sought to investigate the function of CNX in trophoblast differentiation. We created stable CNX-knockdown cell lines by transfecting BeWo cells with a CNX-shRNA expression vector. As Figure 1b illustrates, we successfully established two CNX-knockdown cell lines whose expression of CNX protein was significantly down-regulated (CNX-shRNA-1 and CNX-shRNA-2) compared with expression in non-transfected parental BeWo cells and cells transfected with a vector containing control shRNA (Control-shRNA). We also used immunoblot analysis to study the expression of other ER-resident chaperones including CRT, binding protein of immunoglobulin (BiP), and protein disulfide isomerase (PDI). CRT, BiP, and PDI levels were comparable in all cell groups: parental cells, Control-shRNA cells, and CNX-knockdown cells (Figure 1b).

We then investigated subcellular localization of CNX and PDI by using immunofluorescence microscopy. CNX partially co-localized with PDI in parental and Control-shRNA cells (Figure 1c). Consistent with the results of the immunoblot analysis, we detected no CNX immunoreactive signals in CNX-knockdown cells. Subcellular distributions of PDI, CRT, and BiP were comparable in parental cells, Control-shRNA cells, and CNX-knockdown cells (Figure 1c and Appendix A), which suggested that CNX knockdown had no effect on ER morphology. With phase-contrast microscopy, we observed no notable differences in cell morphology among parental cells, Control-shRNA cells, and CNX-knockdown cells (Figure 1d). Cell proliferation was not affected by CNX knockdown after 24 and 48 h of culture, although growth of CNX-shRNA-1 cells was significantly suppressed at 72 h (Figure 1e).

### 3.2. Effect of CNX Knockdown on FSK-Induced Cell Fusion in BeWo Cells

To investigate the effect of CNX down-regulation on syncytialization, we treated CNX-knockdown cells (CNX-shRNA-1 and CNX-shRNA-2) and control cells (Control-shRNA) with 50 µM FSK for 72 h to induce cell fusion. We then stained the cells with antibodies against ZO-1, a tight junction marker [31]. FSK treatment eliminated ZO-1 proteins at cell-cell borders in Control-shRNA cells, which indicated that cell fusion had occurred (Figure 2a). In CNX-knockdown cells, ZO-1 proteins at cell-cell borders remained after 72 h of FSK treatment, which suggested that CNX-knockdown cells failed to fuse with each other. Quantification of fusion indexes revealed that cell fusion was significantly reduced in CNX-knockdown cells, by approximately 60% (Figure 2b).

### 3.3. Effect of CNX Knockdown on Induction of β-hCG by FSK in BeWo Cells

We next treated Control-shRNA cells and CNX-knockdown cells with 50 µM FSK for 48–72 h, and we then studied the expression of β-hCG, a marker of syncytialization [4,32], at protein and mRNA levels. Figure 3a shows that after a 72 h treatment with FSK, glycosylated β-hCG (25–30 kDa) and non-glycosylated or partially glycosylated β-hCG (20 kDa) were detected in the conditioned medium and lysate of Control-shRNA cells, which indicated that these cells underwent syncytialization. In CNX-knockdown cells, however, FSK-induced up-regulation and secretion of β-hCG were reduced by 60–90%, which suggested that CNX-knockdown cells failed to undergo syncytialization. Results of immunofluorescence analysis with an anti-β-hCG antibody supported immunoblot analysis findings (Figure 3b). Next, we used RT-qPCR analysis to analyze transcriptional expression of β-hCG. Figure 3c indicates that after 24 or 48 h of FSK treatment, the mRNA level of β-hCG was significantly up-regulated in Control-shRNA cells. However, FSK-induced β-hCG up-regulation was significantly reduced in CNX-knockdown cells by 80–90% compared with Control-shRNA cells.

### 3.4. Effect of CNX Knockdown on LHCGR Expression in BeWo Cells

During placental development, β-hCG expression is up-regulated via the binding of β-hCG to the β-hCG receptor LHCGR in an autocrine-dependent manner [3,4]. Indeed, defective STB formation was observed in normal CTB cultures by using LHCGR siRNA, which resulted in a lower hCG secretion [5]. This study also supported the autocrine loop of β-hCG induction and syncytialization in trophoblasts. LHCGR is a seven-transmembrane glycoprotein receptor [33], and CNX is reportedly involved in the maturation and/or membrane localization of LHCGR [34,35]. In the present study, we also found that CNX was included in the immunoprecipitated samples, which were prepared from the lysate of BeWo cells by using the anti-LHCGR antibody (Appendix A). Thus, we hypothesized that CNX-knockdown cells would fail to express LHCGR at the plasma membrane, which may lead to failure of autocrine up-regulation of β-hCG. Figure 4a indicates that CNX knockdown had no effect on cellular LHCGR levels in BeWo cells. Next, to assess the level of LHCGR at the cell surface, we biotinylated membrane proteins and collected them by using avidin beads, and we analyzed LHCGR protein levels in membrane protein fractions by using immunoblots and the anti-LHCGR antibody. The cell-surface LHCGR level decreased 25% in CNX-shRNA-2 cells compared with Control-shRNA cells (Figure 4b). We also analyzed cell-surface expression of LHCGR by means of immunofluorescence microscopy with the antibody against the extracellular N-terminal domain of LHCGR without membrane permeabilization. As seen in Figure 4c, LHCGR signals were detected in Control-shRNA cells but were reduced in CNX-shRNA-2 cells. All these results indicate that CNX down-regulation suppressed cell-surface expression of LHCGR in BeWo cells.

### 3.5. Effects of CNX Knockdown on the Expression of β-hCG in BeWo Cells Induced by the Conditioned Medium Containing hCG

We next investigated whether the autocrine loop regulation of β-hCG was compromised in CNX-knockdown cells. We cultured BeWo cells with or without FSK (50 µM) for 48 h, after which conditioned media were collected and FSK was removed by using an Amicon Ultra centrifugal filter (10-kDa cutoff) and used as β-hCG-containing “FSK-treated CM”. Secretion of β-hCG by FSK-treatment was confirmed by means of immunoblot analysis using the anti-β-hCG (Figure 5a). To investigate whether CNX knockdown affects hCG-induced expression of β-hCG, we treated Control-shRNA and CNX-shRNA-2 cells with FSK-treated CM or Untreated CK (control) for 48 h. Then, the expression of β-hCG mRNA was examined by RT-qPCR in the cells. In Figure 5b, enhanced β-hCG mRNA expression induced by FSK-treated CM was significantly reduced in CNX-shRNA-2 cells compared with Control-shRNA cells.

## 4. Discussion

During gestation, placental cells produce a number of proteins and polypeptides including hormones, cytokines, growth factors, and their receptors, all of which are required to maintain a pregnancy [36]. In general, secretory and membrane proteins are properly folded and assembled in the ER via the protein quality control system with the help of molecular chaperones including CNX, CRT, BiP, and Grp94 [9,10]. Therefore, impairment of the ER quality control system in the placenta is believed to be one cause of defective placental functions that may lead to PE and FGR in humans [11,12].

We previously reported that deficiency of CRT, an ER-resident molecular chaperone, prevented syncytialization of CTB model BeWo cells [20]. However, the effects of depletion of CNX, another ER-resident homologous protein of CRT, on syncytialization have not been investigated. We hypothesized that CNX in CTBs may function in placentation, and we thus studied the role of CNX in syncytialization by taking advantage of shRNA-mediated gene silencing of CNX in BeWo cells. We found that CNX knockdown reduced FSK-induced cell fusion and β-hCG induction in BeWo cells, which suggested that syncytialization-related signaling in villous trophoblast cells was disrupted in the absence of CNX. In placental development, β-hCG production is a pivotal step at the start of development, and β-hCG expression is strictly controlled during gestation [37]. During trophoblast syncytialization, β-hCG promotes differentiation of villous CTBs into STBs in an autocrine-dependent manner via binding of β-hCG to its receptor LHCGR [3,4]. Indeed, we observed that transcriptional expression of β-hCG was significantly upregulated in control BeWo cells that were treated with β-hCG-containing conditioned medium, which supports that autocrine-dependent induction of β-hCG actually occurred in these cells (Figure 5). However, the induction of β-hCG mRNA was significantly suppressed in the CNX-knockdown BeWo cells treated with the β-hCG-containing medium. Collectively, these results strongly suggest that CNX plays a critical role in the regulation of the autocrine loop of β-hCG. In our previous study with CRT-knockdown BeWo cells, FSK-induced cell fusion was suppressed via a reduction in cell-cell interactions, which was due to decreased cell-surface E-cadherin expression [20]. In the current study, expression levels and intracellular localization of E-cadherin were unchanged in CNX-knockdown BeWo cells (Appendix A). These data suggest that CRT and CNX are required for, but contribute in different ways, to CTB syncytialization, i.e., CRT regulates membrane localization of E-cadherin and cell fusion, but CNX regulates membrane expression of LHCGR without affecting the subcellular localization of E-cadherin. A soluble form of 51-kDa CNX was identified in the ER-soluble fraction of human placental tissues [38]. Although we did not detect the 51-kDa form of CNX in BeWo cells here, knowing whether soluble CNX can share chaperone functions with CRT, which is a soluble paralog of CNX, in the ER of placental tissues may be worthwhile.

LHCGR, a member of the G-protein coupling receptor family [33], is a seven-transmembrane glycoprotein receptor located at the plasma membrane in various hCG-responsive human tissues or organs including the testes, brain, ovaries, adipose tissue, adrenal gland, and thyroid gland (Human Protein Atlas) [39]. The immature form of LHCGR was previously reported to be associated with CNX in HEK293 cells expressing rat LHCGR cDNA [40]. The binding of CNX with immature human LHCGR was also confirmed [34]. In that study, the intracellular association between CNX and LHCGR was detected even in the presence of the α-glucosidase inhibitor castanospermine, which disrupts formation of monoglucosylated *N*-glycans for CNX recognition; this finding suggests that CNX may interact with LHCGR in a glycan-independent manner. Also noteworthy is that Davis et al. reported that *N*-glycans of LHCGR were not required for protein folding and cell-surface expression of LHCGR [41], although an earlier work showed the importance of certain *N*-glycans of LHCGR for proper protein folding and function [35]. Several contradictory results concerning the contribution of *N*-glycans in the maturation process of LHCGR have been published, but these studies mostly supported the finding that CNX can function in the maturation and quality control of LHCGR. Thus, our current results suggested that CNX may be required for the proper protein folding and/or sorting of LHCGR to the cell surface in BeWo cells. Clarification of the precise mechanism of how CNX is involved in membrane localization of LHCGR is a future challenge.

In Appendix A, we found CNX expression in human placental tissues from the third trimester and significantly reduced CNX levels in PE placentas, which suggested that impaired placental development may be caused by decreases in placental CNX. Our immunohistochemical analysis of placentas indicated that villous STBs as well as CTBs strongly expressed CNX. We also confirmed that FSK treatment had no effect on CNX protein levels in Control-shRNA and CNX-shRNA-2 cells (Appendix A). Because differentiation of CTBs into STBs is critical for successful placental development and fetal growth [42], these results strongly suggested critical roles of CNX in formation of the STB layer. In addition, we also observed decreased CNX expression in some, but not all, of the tissue samples in PE patients (Appendix A). Given that our cell-based assays clearly showed that downregulation of CNX in CTB model BeWo cells resulted in defects in syncytialization, our immunoblot analysis of PE placentas strongly supports a role of CNX downregulation in PE pathophysiology at least in patients with decreased CNX expression. Clarification of the precise mechanism of the downregulation of CNX in PE placentas is an important future challenge. Differentiation of CTBs to STBs mainly occur in the early phase of the pregnancy. Thus, analyzing the CNX levels and β-hCG levels in the first-trimester placentas will provide strong support for the role of CNX in the STB formation. However, currently, we only have access to the third-trimester placental tissues, but not to the first-trimester tissues, because of ethical limitations in Japan. Further studies to elucidate the role of CNX in the differentiation of CTBs to STBs by using primary cultured villous trophoblasts clearly deserves future investigation. Comparing CNX protein levels in primary cultured villous trophoblasts from normal placentas and PE placentas is an important future focus.

Down-regulation of CNX expression was reported in several human diseases related to metabolic disorders and cancer. CNX expression was reportedly suppressed by cholestasis-induced ER stress, which led to down-regulation of the bile acid importer Na^+^-taurocholate cotransporter protein (NTCP) in liver cells [43]. The decrease in NTCP in the liver is likely a pathogenic factor in cholestasis. ER stress was also implicated in the pathology of non-alcoholic fatty liver diseases [44]. In liver samples from patients with non-alcoholic steatohepatitis, the transcriptional level of CNX was down-regulated, and the level of fully glycosylated functional NTCP was also reduced [45]. These studies suggest that CNX down-regulation may be a common pathogenic mechanism of ER stress-related metabolic disorders or diseases. In addition, CNX is a key factor in major histocompatibility complex class I antigen processing in immunity [46]. In malignant tumor tissues, a decrease in CNX is believed to cause major histocompatibility complex disassembly and thus impaired tumor-antigen presentation, which results in tolerance of tumors to CD8^+^ T-cell-based immune suppression [47]. In human melanoma, reductions in CNX were detected especially in metastatic regions of tissues [48], and these reduced levels may be linked to enhancement of tumor malignancy related to the acquisition of immune tolerance by tumors. Thus, elucidating the detailed mechanism of CNX down-regulation in preeclamptic placentas is another future topic of interest.

## 5. Conclusions

We provide here evidence that CNX regulates FSK-induced CTB syncytialization involving cell fusion and β-hCG secretion. CNX was critical for membrane localization of the β-hCG receptor LHCGR, which is required for up-regulation of the autocrine loop of β-hCG. Our results thus shed light on a novel pathogenic role of CNX in diseases that are associated with placental dysfunction. One study reported that the CNX level in STB extracellular vesicles purified from PE STBs was up-regulated compared with levels in vesicles purified from normal STBs [49], although the pathological relevance of this up-regulation is unclear. Elucidation of therapeutic and preventive potentials of CNX in PE clearly deserves additional investigation.

## Figures and Tables

**Figure 1 biomolecules-12-01050-f001:**
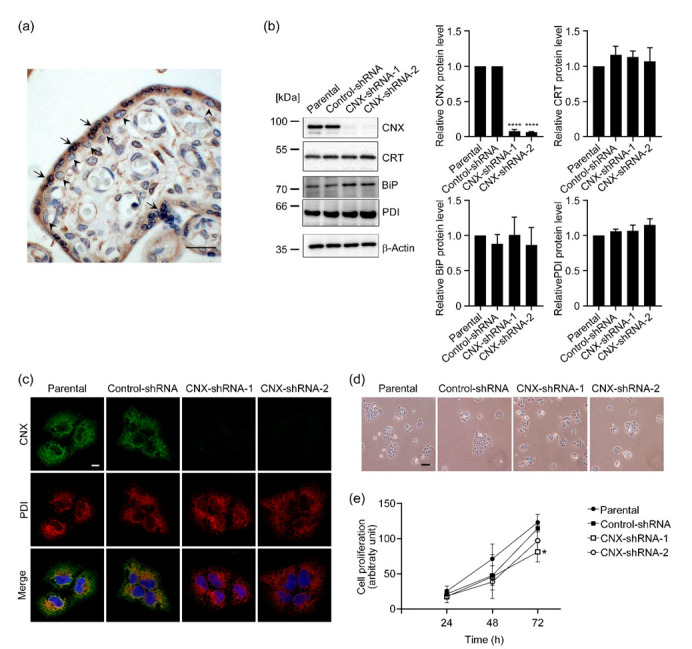
Stable knockdown of CNX expression in BeWo cells. (**a**) Representative section of third-trimester normal placenta was subjected to immunohistochemistry with anti-CNX antibody. STB cytoplasm and CTB cytoplasm were positive for CNX. Arrows show CNX-positive STBs and arrowheads show CNX-positive CTBs. Scale bar, 20 µm. (**b**) We investigated the levels of the ER-resident chaperones CNX, CRT, BiP, and PDI by means of immunoblot analysis with specific antibodies in parental cells, mock-transfected cells (Control-shRNA), and CNX-knockdown cells (CNX-shRNA-1 and CNX-shRNA-2). β-Actin served as the loading control. The graphs show quantification of the levels of each chaperone. Data are shown as means ± SD of three independent experiments. **** *p* < 0.0001. (**c**) Intracellular localization of CNX and PDI was studied in parental cells, Control-shRNA cells, and CNX-knockdown cells by using immunofluorescence microscopy with specific antibodies. PDI staining was used to show the ER. Scale bar: 20 µm. (**d**) Cell morphology was examined by using phase-contrast microscopy. Scale bar: 100 µm. (**e**) Cell proliferation was estimated photometrically in parental cells, Control-shRNA cells, and CNX-knockdown cells by means of crystal violet staining as described in Materials and Methods. Data are shown as means ± SD of three independent experiments. * *p* < 0.05 vs. Control-shRNA cells (72 h).

**Figure 2 biomolecules-12-01050-f002:**
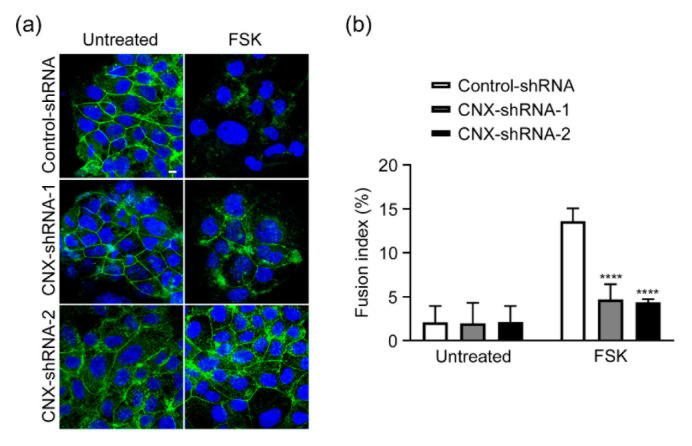
CNX knockdown reduced FSK-induced cell fusion in BeWo cells. (**a**) Control-shRNA cells and CNX-knockdown cells (CNX-shRNA-1 and CNX-shRNA-2) were treated with FSK (50 µM) for 72 h and were then stained with an anti-ZO-1 antibody. Nuclei were stained with DAPI. Scale bar: 20 µm. (**b**) Control-shRNA cells and CNX-knockdown cells (CNX-shRNA-1 and CNX-shRNA-2) were treated with FSK (50 µM) or DMSO (0.05%) for 72 h, after which cell fusion was evaluated and quantified via the fusion index, as described in Materials and Methods. Data are shown as means ± SD of three independent experiments. **** *p* < 0.0001 vs. Control-shRNA (FSK).

**Figure 3 biomolecules-12-01050-f003:**
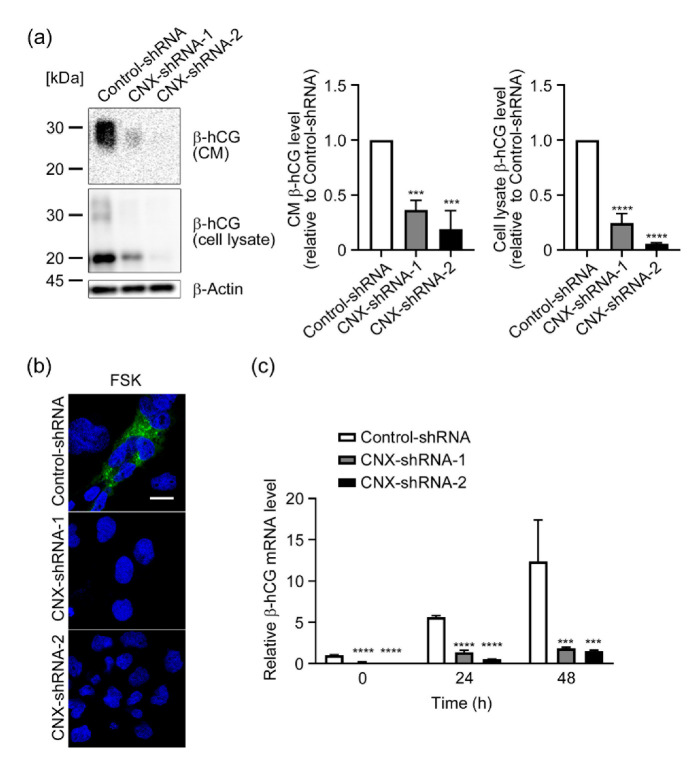
CNX knockdown attenuated FSK-induced β-hCG production in BeWo cells. (**a**) Control-shRNA cells and CNX-knockdown cells (CNX-shRNA-1 and CNX-shRNA-2) were treated with FSK (50 µM) for 72 h, after which cells and conditioned media (CM) were collected and β-hCG levels were analyzed by means of immunoblotting. The graphs show quantification of β-hCG levels in cell lysate and CM. Data are shown as means ± SD of three independent experiments. *** *p* < 0.001; **** *p* < 0.0001 vs. Control-shRNA. (**b**) Control-shRNA cells and CNX-knockdown cells (CNX-shRNA-1 and CNX-shRNA-2) were treated with FSK (50 µM) for 72 h. Intracellular expression and localization of β-hCG were studied by using immunofluorescence microscopy with specific antibodies. Scale bar: 20 µm. (**c**) Control-shRNA cells and CNX-knockdown cells (CNX-shRNA-1 and CNX-shRNA-2) were treated with FSK (50 µM) for 48 h. Total RNA was collected at the indicated time points, and then RT-qPCR was performed for β-hCG as described in Materials and Methods. Data are shown as means ± SD of three independent experiments. *** *p* < 0.001; **** *p* < 0.0001 vs. Control-shRNA at each time point.

**Figure 4 biomolecules-12-01050-f004:**
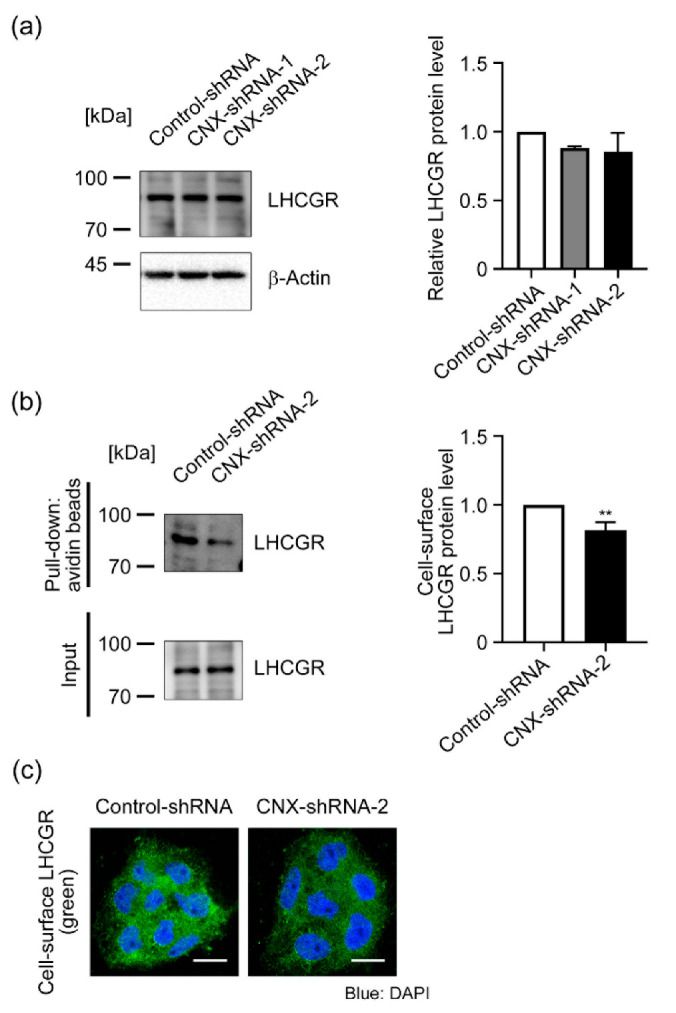
CNX knockdown reduced cell-surface expression of LHCGR in BeWo cells. (**a**) LHCGR expression was analyzed in Control-shRNA, CNX-shRNA-1, and CNX-shRNA-2 cells by means of immunoblot analysis with the specific antibody. The graph shows relative LHCGR levels. Data are shown as means ± SD of three independent experiments. (**b**) Cell-surface proteins were biotinylated in Control-shRNA cells and CNX-shRNA-2 cells, after which biotinylated proteins were collected by using avidin beads. After the avidin beads were washed, biotinylated proteins were eluted from beads and were subjected to immunoblot analysis with the antibody against LHCGR. The graph shows relative cell-surface levels of LHCGR. Data are shown as means ± SD of three independent experiments. ** *p* < 0.01 vs. Control-shRNA. (**c**) Cell-surface expression of LHCGR in Control-shRNA cells and CNX-shRNA-2 cells was investigated by using immunofluorescence microscopy with the antibody recognizing the extracellular N-terminal domain of LHCGR under non-cell-permeable conditions as described in Materials and Methods. Scale bar: 20 µm.

**Figure 5 biomolecules-12-01050-f005:**
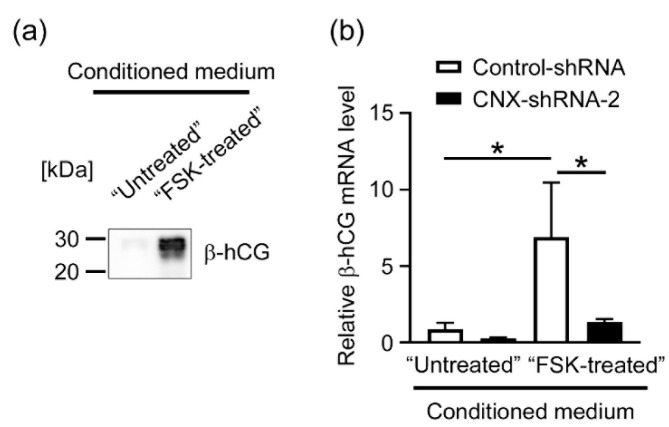
CNX knockdown attenuated autocrine-dependent induction of β-hCG in BeWo cells. BeWo cells-derived conditioned medium samples were obtained as FSK-treated CM (β-hCG-containing medium) and Untreated CM (control medium) as described in Materials and Methods. (**a**) FSK-induced production of β-hCG was examined in the conditioned medium samples by immunoblot analysis using the specific antibody. (**b**) CNX-knockdown cells (CNX-shRNA-2) and control cells (Control-shRNA) were treated for 48 h with FSK-treated CM or Untreated CM as described in Materials and Methods. Then the expression of β-hCG mRNA was examined by RT-qPCR. * *p* < 0.05 vs. FSK-treated Control-shRNA.

## Data Availability

All data is included in the manuscript and/or supporting information.

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
