# Peer review of "Calnexin Is Involved in Forskolin-Induced Syncytialization in Cytotrophoblast Model BeWo Cells"

_biomolecules, 2022, doi:10.3390/biom12081050_

Round 1
Reviewer 1 Report
This manuscript by Matsukawa et al. elucidates the role of the calnexin (CNX) in trophoblast cell line in vitro. CNX knockdown Bewo cells showed reduced fusion and hCG transcripts/protein after forskolin induction. The authors report decreased surface expression of luteinizing hormone/chorionic gonadotropin receptor that suggests the presence of a dysfunctional autocrine loop of hCG up-regulation. Overall, the results are intriguing, but I have some major criticisms which in my opinion should be addressed
The expression of CNX in placenta is mainly in syncytiotrophoblast from the data of Figure 1a and the Human Protein Atlas. CNX expression is slightly or negative in cytotrophoblast cells in the human placenta. The author state that “We hypothesized that CNX in STBs may function in placentation” in line 403 (and in line 453). Also, the author state that Bewo cells were used as CTB like “CTB model Bewo cells” in line 458 (also line 401and Title). Please make sure and explain that the discrepancy of CNX localization in this study. It could be encouraged to the authors to use “CTB model, CTB syncytialization” carefully or add a sentence to the discussion to present the limitations of this experiments using the choriocarcinoma cell line Bewo.
Based on these, the authors should show the expression change of CNX transcripts or protein after forskolin induction in CNX knockdown and control/un-treated Bewo cells. CNX expression is up-regulated with forskolin treatment or …?
Why did you provide sup Fig S4 as a supplementary figure and not an original one, in the manuscript? It should be as an original figure in the manuscript if the author would state these results in the abstract. In addition, I recommend to shows the hCG expression in these PE samples. Would it be too much to ask for if the authors examined whether hCG expression is also decreased in these clinical PE samples. Is it not easy to show hCG protein expression using clinical sample lysate (or hCG transcripts)? Please leave it as supplementary figure if hCG data are not available.
(Minor)
・ Title inappropriate at that point.
・ Respective IgG controls should be presented in figure 1a including the scale bar.
・ It should be better to show the antibody catalog number in the Methods section.
・ Line 481-484 need to remove. No need previous reviewer’s comments.
・ Line 425-426, the author state that “CNX regulates membrane expression of LHCGR without affecting cell fusion”. This sentence is somehow confusing. CNX is not associate with cell fusion?
Author Response
Author Reply
Reviewer 1
This manuscript by Matsukawa et al. elucidates the role of the calnexin (CNX) in trophoblast cell line in vitro. CNX knockdown Bewo cells showed reduced fusion and hCG transcripts/protein after forskolin induction. The authors report decreased surface expression of luteinizing hormone/chorionic gonadotropin receptor that suggests the presence of a dysfunctional autocrine loop of hCG up-regulation. Overall, the results are intriguing, but I have some major criticisms which in my opinion should be addressed
The expression of CNX in placenta is mainly in syncytiotrophoblast from the data of Figure 1a and the Human Protein Atlas. CNX expression is slightly or negative in cytotrophoblast cells in the human placenta. The author state that “We hypothesized that CNX in STBs may function in placentation” in line 403 (and in line 453). Also, the author state that Bewo cells were used as CTB like “CTB model Bewo cells” in line 458 (also line 401and Title). Please make sure and explain that the discrepancy of CNX localization in this study. It could be encouraged to the authors to use “CTB model, CTB syncytialization” carefully or add a sentence to the discussion to present the limitations of this experiments using the choriocarcinoma cell line Bewo.
Response: First, we thank the reviewer for the careful and critical reading of our manuscript and for his/her constructive comments.
We agree with the reviewer that there is a discrepancy between the result of CNX immunohistochemical analysis and the cell model that was used in the previous version. We re-evaluated our immunohistochemical data and noticed that STBs as well as CTBs are strongly positive for CNX. The new data are included as Figure 1a and the legend on page 6, lines 269–271, and Supplemental Figure 1a. Accordingly, we would like to keep the term “the CTB model” in the text and the title. In order to avoid any confusion, we deleted the description regarding the Human Protein Atlas and revised the main text on page 5, lines 239–242; page 11, line 405; and page 12, lines 454–455.
Based on these, the authors should show the expression change of CNX transcripts or protein after forskolin induction in CNX knockdown and control/un-treated Bewo cells. CNX expression is up-regulated with forskolin treatment or …?
Response: We thank the reviewer for his/her constructive comment. We analyzed the effect of syncytialization on CNX protein levels in BeWo cells, and noted no difference caused by FSK treatment. The new result was shown in newly prepared Supplemental Figure S6 and mentioned on page 12, lines 455–456 in the main text.
Why did you provide sup Fig S4 as a supplementary figure and not an original one, in the manuscript? It should be as an original figure in the manuscript if the author would state these results in the abstract. In addition, I recommend to shows the hCG expression in these PE samples. Would it be too much to ask for if the authors examined whether hCG expression is also decreased in these clinical PE samples. Is it not easy to show hCG protein expression using clinical sample lysate (or hCG transcripts)? Please leave it as supplementary figure if hCG data are not available.
Response: We thank the reviewer for the kind suggestion. We challenged to detect β-hCG in placenta samples, but unfortunately, we failed. Because β-hCG blood levels usually achieve a peak between 8 to 12 weeks of pregnancy during which placentation occurs and CTBs extensively differentiate to STBs, but begin to decrease after approximately 24 weeks of pregnancy, which suggests that the third trimester placenta may not produce extensively β-hCG anymore. Thus, it may be difficult to validate the effect of CNX reduction on placental β-hCG production by using the third trimester placentas. Currently, we do not have access to the human placenta samples of the first trimester because of ethical limitations in Japan and in our institute, and we think that further studies to investigate the effect of CNX knockdown on β-hCG production and syncytialization by using primary cultured trophoblasts are necessary. We discussed this point on page 12, lines 465–472 and modified the Abstract on page 1, lines 24–26. Also, please be informed that we kept the CNX immunoblot of placenta samples in Supplemental information as the recommendation by this reviewer.
(Minor)
・ Title inappropriate at that point.
Response: Because we found that CTBs also expressed CNX in the third trimester human placenta, we keep the current title. Please see our response to the comment 1.
・ Respective IgG controls should be presented in figure 1a including the scale bar.
Response: The image of a rabbit isotype IgG control is shown in Supplemental Figure 1b and mentioned on page 5, lines 242–244 in the main text. We also added scale bars in Figure 1a and Supplemental Figure S1.
・ It should be better to show the antibody catalog number in the Methods section.
Response: The catalog numbers of antibodies are shown in the Materials.
・ Line 481-484 need to remove. No need previous reviewer’s comments.
Response: We apologize for our careless mistake. The comment is deleted.
・ Line 425-426, the author state that “CNX regulates membrane expression of LHCGR without affecting cell fusion”. This sentence is somehow confusing. CNX is not associate with cell fusion?
Response: We thank the reviewer for pointing this out. We rephrased the sentence as “but CNX regulates membrane expression of LHCGR without affecting the subcellular localization of E-cadherin” on page 11, line 427.
Reviewer 2 Report
This is a very well-conducted, focused study, and the description of the methods, presentation of the data, and discussion are all superb. I have virtually no suggestions for improvements apart from the following general points:
1. The sole use of the BeWo cell model is perhaps a negative. Although well-validated and widely used for studies of syncytialisation, ideally some aspects of the study would be replicated in another system. At the least, in the Discussion, the authors might suggest alternative models that could be used in replication studies.
2. The data on expression of CNX in primary term placental tissues (normal and PE) is hidden away in a Supplementary file. This is an important figure and I think it could be included in the main text? I also consider that it would have been good to include analysis of expression in first trimester, although that material may not have been available. In any case, a more extensive comparison of CNX expression in normal and pathological trophoblast would be a good addition to buttress the clinical relevance of the BeWo derived data.
Author Response
Author Reply
Reviewer 2
This is a very well-conducted, focused study, and the description of the methods, presentation of the data, and discussion are all superb. I have virtually no suggestions for improvements apart from the following general points:
- The sole use of the BeWo cell model is perhaps a negative. Although well-validated and widely used for studies of syncytialisation, ideally some aspects of the study would be replicated in another system. At the least, in the Discussion, the authors might suggest alternative models that could be used in replication studies.
Response: We first thank the reviewer for the constructive and scientifically outstanding review of our manuscript. We truly appreciate that the reviewer pointed out several critical issues.
We agree with the reviewer that additional analyses by using different assay systems would strengthen our results. We discussed this point in the Discussion on page 12, lines 472–473.
- The data on expression of CNX in primary term placental tissues (normal and PE) is hidden away in a Supplementary file. This is an important figure and I think it could be included in the main text? I also consider that it would have been good to include analysis of expression in first trimester, although that material may not have been available. In any case, a more extensive comparison of CNX expression in normal and pathological trophoblast would be a good addition to buttress the clinical relevance of the BeWo derived data.
Response: We agree with the reviewer that analyzing the first trimester placentas is important for supporting our claim that CNX in CTBs is critical for differentiation into STBs, because we used BeWo cells as the CTB model and CTBs extensively differentiate to STBs during the first trimester of pregnancy. We also noticed that although our immunoblot in Supplemental Figure S5 showed decreased CNX in the PE placentas, the immunoblot cannot distinguish CNX that is expressed in CTBs from CNX expressed in other types of cells. Thus, as the reviewer pointed out, we think that additional analyses such as those with primary cultured trophoblasts obtained from normal placentas and PE placentas are important future challenge. We mentioned this on page 12, lines 465–472 and modified the Abstract on page 1, lines 24–26. Please be informed that the CNX immunoblot is still included in Supplemental Information per request by the Reviewer 1.